# Epigenetics, Microbiome and Personalized Medicine: Focus on Kidney Disease

**DOI:** 10.3390/ijms25168592

**Published:** 2024-08-06

**Authors:** Giuseppe Gigliotti, Rashmi Joshi, Anam Khalid, David Widmer, Mariarosaria Boccellino, Davide Viggiano

**Affiliations:** 1Department Nephrology and Dialysis, Eboli Hospital, 84025 Eboli, Italy; g.gigliotti@aslsalerno.it; 2Department Translational Medical Sciences, University of Campania, 81100 Naples, Italy; rashmi.joshi@unicampania.it (R.J.); anam.khalid@unicampania.it (A.K.); davide.viggiano@unicampania.it (D.V.); 3Vidmar-Daj Consulting, New York, NY 07093, USA; dajwidmer@gmail.com; 4Department Experimental Medicine, University of Campania, 81100 Naples, Italy; 5Department Life Sciences, Health and Health Professions, Link University, 00165 Rome, Italy

**Keywords:** microbiome, IgAN, epigenome

## Abstract

Personalized medicine, which involves modifying treatment strategies/drug dosages based on massive laboratory/imaging data, faces large statistical and study design problems. The authors believe that the use of continuous multidimensional data, such as those regarding gut microbiota, or binary multidimensional systems properly transformed into a continuous variable, such as the epigenetic clock, offer an advantageous scenario for the design of trials of personalized medicine. We will discuss examples focusing on kidney diseases, specifically on IgA nephropathy. While gut dysbiosis can provide a treatment strategy to restore the standard gut microbiota using probiotics, transforming epigenetic omics data into epigenetic clocks offers a promising tool for personalized acute and chronic kidney disease care. Epigenetic clocks involve a complex transformation of DNA methylome data into estimated biological age. These clocks can identify people at high risk of developing kidney problems even before symptoms appear. Some of the effects of both the epigenetic clock and microbiota on kidney diseases seem to be mediated by endothelial dysfunction. These “big data” (epigenetic clocks and microbiota) can help tailor treatment plans by pinpointing patients likely to experience rapid declines or those who might not need overly aggressive therapies.

## 1. Introduction

Often, the strategist reasons that when facing a situation similar to a past one, a past strategy can be applied if that one was successful in the past. However, conditions never repeat precisely as they occurred in the past, and history—like the weather—displays chaotic behavior and is highly sensitive to even the slightest change of state, and thus has an unpredictable future in the middle to long run.

In medicine, we apply the same fallacy: one pill with the same dosage for everybody, with a supposedly similar basal condition. Cardioaspirin is applied at the same dosage for everybody, and when using antihypertensives or hypocholesterolemic drugs, the same target blood pressure or cholesterol levels must be used for everybody, irrespective of gender, age, genetics, epigenetics, microbiota, or other conditions.

Certainly, this depends on a limitation in our reasoning abilities and the necessity of implementing relatively simple methods when interpreting the fuzzy and overwhelming amount of medical data available in the literature.

As soon as we increase the amount of information a physician is required to consider before using a treatment, it becomes evident that no patient has the same condition as any other. The physician then faces the paralyzing situation of this specific patient having no clearly described treatment algorithm to use. Therefore, the physician can only try to use the same remedy he/she successfully used in a case considered close enough to the present one.

Dealing with “personalized medicine”, which hopes to guide treatment based on a vast amount of information gathered from the patient, also means that there are as many treatment conditions as there are patients, and this represents a major challenge: how to demonstrate that a specific treatment would be the best for a specific, unique condition.

One relatively simple example is the large amount of antibiotics available, each one with many possible dosages and an infinite possibility regarding their combination. The standard guide nowadays is to use the “antibiogram”, thus excluding several choices, and then selecting the best antibiotic based on relatively few rules. It does work, but maybe we are not using the optimal solution.

The statistical and theoretical problems of this approach are enormous. Here, we will use the term “state” to refer to the set of clinical/laboratory data used to define a diseased patient. In the remaining manuscript, we will treat possible states as a continuum rather than a discontinuous function of a multidimensional “state space”. For example, since we will focus on the microbiome and epigenetic clock, the epigenetic age of a subject and the amount of each of the bacteria are continuous variables that identify a single point in multidimensional space. 

We believe that the only hope for personalized medicine is that the dose–state response for a specific drug has a continuous and smooth representation. Therefore, not all possible states must be studied to derive the dose–state–response hypersurface.

Recent advances in biomedical research have revealed a fascinating and complex relationship between epigenetics and the gut microbiota, two fields that were once considered distinct but are now recognized as profoundly interconnected. This intricate interplay has significant implications for human health and disease, opening new avenues for therapeutic interventions and personalized medicine.

Epigenetics, the study of heritable changes in gene expression that do not involve alterations to the underlying DNA sequence, has emerged as a critical factor in understanding how environmental influences can affect gene activity. Concurrently, the microbiota, particularly the diverse community of microorganisms residing in the gut, has been recognized as a major player in human physiology, influencing everything from digestion and immunity to brain function.

The bidirectional communication between epigenetics and the microbiota forms a dynamic system that shapes various aspects of human health. On one hand, the microbiota represent an environmental factor that influences epigenetic modifications. Bacterial metabolites, such as short-chain fatty acids (SCFAs), have been shown to affect epigenetic markers like DNA methylation and histone acetylation directly. These modifications can, in turn, alter gene expression patterns in host cells, potentially influencing susceptibility to various diseases.

Conversely, epigenetic mechanisms can significantly impact the composition and function of the gut microbiota. For instance, epigenetic modifications can modulate the expression of genes involved in maintaining the intestinal barrier function, thereby influencing the types of bacteria that colonize the gut. This reciprocal relationship creates a complex feedback loop, where changes in one system can cascade through the other, affecting overall health status.

The implications of this interplay extend to various health conditions and diseases. In the context of gut health, dysbiosis—an imbalance in the gut microbiota—has been linked to inflammatory diseases such as inflammatory bowel disease (IBD) and irritable bowel syndrome (IBS). Epigenetic changes influenced by the microbiota may contribute to the development and progression of these conditions. Similarly, in metabolic disorders like obesity and type 2 diabetes, the gut microbiota’s role in energy metabolism interacts with epigenetic modifications, potentially influencing disease onset and progression.

Furthermore, the gut–brain axis, a complex network of bidirectional communication between the gut and the brain, exemplifies how the microbiota can influence brain function through epigenetic mechanisms. This interaction impacts mental health, impacting mood, behavior, and cognitive function.

This is a narrative review of the combined use of microbiota and epigenetics as tools in personalized medicine. Because the literature is scant, it was impossible to perform a rigorous analysis of the literature as is performed in meta-analysis works, which will be possible only when larger amounts of clinical data are available. All this notwithstanding, the new area of research shows exciting potential as regards its future therapeutic impact, and the authors believe that having a starting review will be of great help to the clinical and scientific community

## 2. Personalized Medicine in Kidney Diseases Based on Gut Microbiota Data

Precision or personalized medicine aims at modulating medical management based on a large dataset pertaining to each patient, such as genomic, environmental, imaging or microbiota data [1].

Among the fields of personalized medicine is the study of how the gut microbiome (community of microorganisms in the gut) and nutrition alter the response to disease [2]. Indeed, the microbiome plays a role in various diseases, including kidney disease [3].

The microbiome consists of 10–100 trillion symbiotic microbial cells (bacteria, fungi) that inhabit the gut [4]. These play roles in functions such as carbohydrate digestion and immune function [5,6]. The composition and function of the microbiome in early life can influence long-term health and disease development such as asthma, obesity, brain and muscle disorders [7,8]. 

Due to the large interindividual variability of the genome, and its relevance for human disease, it is tempting to attribute the variability of human diseases to the composition of the microbiota. The link between the microbiota and disease, however, is not clear.

A major hypothesis is that the gut microbiota modify the reactivity of the immune system, thereby triggering non-adaptive responses that, at the end, damage various organs.

Here, we have analyzed the role of the microbiota in three autoimmune diseases that have been linked to changes in the microbiota: inflammatory bowel disease (IBD) and specifically Crohn’s disease, Systemic Lupus Erythematosus (SLE) and IgA nephropathy (IgAN). These conditions are very different from the point of view of the organ involved: IBD involves primarily the digestive system; IgAN involves almost exclusively the kidney, and SLE is a systemic disease affecting small blood vessels in multiple organs. IBD shares with IgAN some pathogenic mechanisms and a therapeutic approach (the use of gut-released budenoside). SLE has also some similarity with IgAN, as in both cases a deposition of IgA antibodies can be observed in kidney glomeruli, with the development of progressive fibrosis (see also Figure 1). The three diseases also share a pattern of evolution over time in acute worsening phases or “flares”.

In Table 1, we report current evidence relating to the development of an immune disease (specifically Crohn’s disease, Systemic Lupus Erythematosus (SLE) and IgA nephropathy (IgAN)) and the imbalance of a specific bacterium.

Indeed, an increased amount of gut enterobacteria has been linked to both IgAN and Crohn’s disease [9,10]. Proteobacteria have been linked to the development of both SLE and Crohn’s disease [11,12], whereas Firmicutes is protective in these conditions [11,12,13]. Therefore, specific bacteria in the gut may trigger autoimmune diseases, and information stored within microbiomes may help us detect at-risk subjects. What is attractive, though, is the possibility that the introduction of “protective” bacterial species or a reduction in “negative” bacteria may reduce the triggering of these autoimmune diseases. In this regard, the analysis of gut microbiota may open up the possibility of a precision medicine approach [16].

IgA nephropathy (IgAN) is a chronic kidney disease characterized by the deposition of IgA antibodies in the glomeruli, leading to inflammation and progressive renal damage [17]. The exact etiology of IgAN is not fully understood, but recent research suggests that the gut microbiota may play a significant role [18]. Accordingly, a major hypothesis is that aberrant immune responses to gut mucosal antigens might foster the production of an IgA variant that, when released in the bloodstream, is entrapped among the mesangial cells of the kidney glomeruli. Obviously, this hypothesis supports the view of a change in the gut microbiota as a major contribution to the stimulation of the gut immune system (see Figure 1) [19].

An imbalance in the gut microbiota or dysbiosis would alter mucosal immunity and cause the overproduction of aberrantly glycosylated IgA, which may then deposit in the glomeruli.

The mechanism linking gut dysbiosis and aberrant IgA production is, however, not well defined. It has been hypothesized that specifically “toxic” bacteria may disrupt the gut barrier and gap junctions, which are very relevant in many organs, and thus increase intestinal permeability, allowing for the adsorption of “pro-inflammatory” microbial products. It is also possible that specific bacteria induce a chronic gut inflammatory state, as in IBD [20].

IBD encompasses Crohn’s disease and ulcerative colitis, both of which involve chronic inflammation in the gastrointestinal tract. Being an inflammatory state in the digestive system, this group of diseases is strongly suspected to be driven by gut dysbiosis [21]. Some interventions support this hypothesis: (1) some probiotics (living “protective” microorganisms ingested as foods or supplements) have shown promise when used in IBD, particularly strains like Lactobacillus and Bifidobacterium [22]; (2) fecal microbiota transplantation (the transfer of fecal bacteria from a healthy donor to a patient) can induce IBD remission [23]; (3) FODMAP diet and exclusive enteral nutrition have been effective in IBD [24]; (4) antibiotics that can disrupt gut microbiota may reduce inflammation in IBD [25].

Overall, analyses of gut microbiota and the consequent undertaking of interventions are expected to modify the triggering of IgA nephropathy. However, the distinction between an aggressive disease that will progress towards fibrosis and dialysis and a “benign” form that responds to current treatments is unlikely to be explained by the microbiota, and may rather require a different approach: epigenetic modifications.

### IgA Nephropathy and Microbiota

In recent years, the complex interplay between IgA nephropathy (IgAN), gut microbiota, and epigenetic aging has emerged as a promising area of research. IgAN, characterized by IgA immune complex deposition in the kidneys, appears to be influenced by gut dysbiosis, which can disrupt intestinal permeability and trigger abnormal immune responses. The gut microbiome’s role in modulating immune function and producing beneficial metabolites, such as short-chain fatty acids, suggests potential therapeutic avenues for IgAN management. Concurrently, epigenetic aging, measured by DNA methylation patterns, may offer insights into disease progression and severity in IgAN patients. Chronic inflammation, a hallmark of IgAN, can induce epigenetic modifications that may accelerate biological aging. Environmental factors, including diet and lifestyle, further complicate this relationship by influencing the microbiome and epigenome. As our understanding of these interconnected processes grows, novel approaches to IgAN treatment are emerging. These include microbiome modulation strategies, developing epigenetic biomarkers for improved diagnosis and prognosis, and exploring anti-aging interventions to mitigate disease progression. Ultimately, a personalized medicine approach that integrates microbiome profiles, epigenetic measurements, and traditional clinical markers may pave the way for more effective, targeted treatments for IgAN patients.

## 3. The Epigenetic Clock

One major theory of aging is that harmful substances damage cells, contributing to their aging process. A typical example states that free radicals, produced during cellular metabolism, induce molecular damages, which accumulate and damage cells, contributing to their dysfunction and, at the system level, to aging. This theory proposes that the varying lifespans of different cell types are not always directly related to the overall lifespan of the organism [26].

Additionally, there’s the well-known concept called the Hayflick limit, proposing that cells can only divide a certain number of times before they stop. However, recent research has raised doubts about whether this limit applies universally to all cells [27]. 

Another theory revolves around telomeres, which are protective caps on the ends of DNA strands. It suggests that with each cell division, telomeres get shorter, and this process is linked to aging [28]. Despite these theories offering insights, there are unresolved issues, such as inconsistencies and challenges in explaining aging at both cellular and organismal levels. Aging is a complex, multifactorial process, and no single theory can fully account for all its aspects. All this notwithstanding, a recent big advancement in cellular ageing has been represented by the discovery of the “epigenetic clock”. Epigenetic variations are chromosomally inherited changes in gene function, but they do not involve alterations to the underlying DNA sequence. These changes can influence gene expression, determining the switching on and off of the gene without affecting the genetic code itself. They are chemical modifications of DNA and associated proteins such as histones that include DNA methylation, histone modification and the activity of non-coding RNAs [29]. These modifications play a crucial role in regulating various biological processes. This includes how organisms develop, how cells specialize, and how bodies respond to their environment. Epigenetic changes can also influence a person’s risk of developing diseases and might offer clues for new treatments. For instance, epigenetics might explain why identical twins, who share the same DNA (genotype), can have different physical traits (phenotype) [30]. Based on the analysis of these epigenetic markers, estimating an individual’s biological age is referred to as the epigenetic clock. Indeed, certain epigenetic modifications such as DNA methylation patterns at specific gene locations show a correlation with age.

Researchers have identified sets of DNA methylation sites, often CpG dinucleotides, that exhibit predictable changes as an individual ages. In the case of mammals, both hypo- and hypermethylation are associated with ageing. A gradual reduction in overall methylation was observed both during the in vitro cultivation of fibroblasts [31] and in ageing animals [32,33]. Hypomethylation is generally observed in repetitive sequences, for example, in transposable elements [34,35], whereas hypermethylation is observed in specific genes of ageing individuals. For example, the methylation of CpG islands is linked to various genes, such as the one that encodes estrogen receptors [36]. Since aging is accompanied by a progressive reduction in methylation in DNA “gene-free” regions and progressively increased methylation in regions linked to various genes, one may think of this clock as a movement of methylation sites from one DNA site to another. Apparently, no specific gene is relevant for this clock: different algorithms that use DNA methylation to estimate chronological age rely on very different genes, even though they perform equally well [37]. In other words, the global DNA methylation (or maybe the ratio of methylation gene-free versus GPG islands) is more important to determining the age than which DNA region becomes progressively methylated, as if each methylation event acts in an additive manner. In line with this idea, it is possible to determine the age even from just the ribosomal DNA methylation state [38]. 

Epigenetic clocks have emerged as groundbreaking tools in the realm of ageing research, offering a window into the molecular intricacies that govern the biological ageing process. Two well-known examples of the epigenetic clock are the Horvath clock and the Hannum clock. These clocks use a combination of specific DNA methylation sites and sophisticated statistical models to estimate biological age accurately. The resulting biological age can then be compared to the person’s chronological age to assess whether they appear to be ageing faster or slower than expected.

Epigenetic clocks have been applied in various research areas including the study of ageing disease susceptibility, and the impact of lifestyle factors on the ageing process. They provide a valuable tool for understanding the molecular basis of ageing and may have implications for personalized medicine and interventions to promote healthy ageing. The precise nature of what DNA methylation age signifies is not fully understood yet. Horvath proposed that it reflects the cumulative impact of an epigenetic maintenance system, although the specifics remain unclear. The correlation between the DNA methylation age of blood and later life all-cause mortality has been used to suggest a connection to ageing processes [39,40,41]. However, if a specific CpG site is directly causative in ageing, its mortality impact would reduce its likelihood of being observed in older individuals, making it less likely to be selected as a predictor. Therefore, the 353 CpGs clocks likely lack a direct causal effect [42], and instead represent an emergent property of the epigenome. These clocks usually require chronological age for the prediction of biological age, which does not seem a very reliable way to confirm that age prediction depends on the methylation pattern completely. Though they are “prediction” methods only, they still provide a deeper insight into the biological age of an individual.

### 3.1. Epigenetic Clock and Kidney Diseases

Epigenetic clocks can be used to identify individuals at high risk of developing kidney disease, even before the onset of symptoms. By assessing a person’s biological age through DNA methylation patterns, epigenetic clocks can pinpoint those with accelerated kidney aging who may be predisposed to renal dysfunction. This information can then guide targeted preventive measures and close monitoring for these high-risk individuals [43,44,45]. The epigenetic clock can help in the identification of many age-related diseases [46,47]. Since the epigenetic clock provides insight into biological age, it can be helpful in identifying the disease, and can guide the treatment decision. Epigenetic clocks, helpful in predicting methylation sites that correlate with kidney function, can be helpful in the early detection of the disease [48]. Studies have also shown that individuals with CKD (chronic kidney disease) and ESRD (End-stage renal disease) exhibit enhanced biological aging, as measured by the epigenetic clock, compared to healthy individuals. These can also be used as biomarkers to check the progression of disease and its severity [45,49]. Epigenetic clocks can also identify disease-specific biomarkers that are differentially expressed in patients with kidney diseases. Biomarkers related to chronic inflammation and immune responses play important roles in the development of kidney diseases such as CKD and ESRD [45,49]. They can be used to determine the levels of acceleration of these diseases by comparing the tissues of kidney disease patients with those of healthy individuals. Advanced CKD patients exhibit increased DNA damage in urinary shedding, which correlates with an accelerated epigenetic clock. These can be used as predictive markers [50]. Epigenetic clocks can also inform treatment decisions for patients with established kidney disease. For example, they may help in identifying individuals who are more likely to progress to kidney failure, allowing clinicians to tailor aggressive treatment plans accordingly. Conversely, epigenetic clocks could identify patients less prone to rapid disease progression, potentially sparing them from overly intensive therapies. In a study, it was found that not dialysis, but kidney transplants partially reduce the accelerated aging seen in CKD patients measured by epigenetic clock [45,50]. By tracking changes in epigenetic age over time, clinicians can use these biomarkers to monitor how well a patient is responding to a given treatment for kidney disease. Epigenetic clocks that fail to improve or even worsen during the course of therapy may signal the need to adjust the treatment approach.

### 3.2. Epigenetic Clock and “Sequential Fibrosis” in the Kidney

The kidney comprises discrete entities, the glomeruli, the numbers of which are proportional to body mass. Glomerular size is the same both in a rat and an elephant, but they are very different in number. As shown in Figure 2, a linear relationship exists between maximum lifespan and the number of glomeruli. 

This data are not conclusive because they may simply reflect the relationship between lifespan and the sizes of different animals. However, the glomeruli undergo a process that we call “sequential fibrosis”. We introduce this neologism, “sequential fibrosis”, to indicate the progressive, physiological, non-simultaneous loss of glomeruli throughout life. With age, the glomeruli undergo a fibrosis process, which can be easily deduced from a reduction in the glomerular filtration rate (eGFR) [55].

Using the MDRD equation
GFR in mL/min per 1.73 m^2^ = 186 × SerumCr ^−1.154^ × age ^−0.203^
where “age” is the age in years and SerumCr is the creatinine in mg/dL, it is easy to estimate the relationship between age and eGFR (shown in Figure 2) [56]. 

Here we have used the MDRD equation rather than other well established estimates of GFR, such as the CKD-EPI, because it shows a smooth, continuous behavior. The MDRD formula is: eGFR = k_1_ × SerumCr ^k2^ × age ^k3^
where SerumCr is serum creatinine, and k_1_, k_2_ and k_3_ are three constants (k_1_ depends on gender and race).

Conversely, the CKD-EPI formula is: eGFR = k1 × min (SCr/k2, 1)^k3^ × max (SCr/k2, 1)^k4^ × K5^Age^

The reader can easily identify the two non-linear functions “min” and “max”, which introduce a non-smooth behavior of the function.

Though the CKD-EPI equation shows an improvement in the estimate of the average eGFR in a population, there is little if any improvement in the variability of the estimates around the mean; the improvement in CKD-EPI regarding the estimate of the true average GFR in a population comes at the cost of the greater complexity and non-linearity of the formula, which might represent a major theoretical limitation in the generalizability of the data to populations not included initially in the CKD-EPI study. These considerations have led us to adopt the MDRD formula in the present study.

The hypothesis is therefore that the kidney is a potential digital biological clock, connected with aging. Several phenomena support this hypothesis. First, people with kidney failure who require hemodialysis show signs of premature aging [57]. Furthermore, individuals with renal failure who receive a kidney transplant have a longer lifespan compared to those who undergo hemodialysis [58]. The brain shows clear signs of premature aging in individuals whose number of glomeruli is lower than normal [59]. Moreover, experiments have shown that the kidneys themselves have an autonomous circadian clock, supporting the idea that the kidney is regulated by the main clock of which we know, that is, the epigenetic clock.

### 3.3. Epigenetic Clock and Endothelial Cells

Endothelial cells become senescent with aging. This seems to be regulated by physical exercise. Endothelial senescence is thought to be linked to overall vascular function, and therefore to be relevant in diseases of the kidney, particularly IgAN [60]. Indeed, endothelial senescence is linked to a reduced number and function of endothelial precursor cells, and therefore endothelial regeneration. The resultant decline in the capacity for tube-formation would result in greater scarring resulting caused by inflammatory diseases such as IgAN [61] (Figure 3).

Indeed, the epigenetic clock is initiated upon the differentiation of human embryonic stem cells (ESCs) into endothelial cells with low genome-wide DNA methylation levels [62]. Therefore, it is possible to estimate gestational age using DNA methylation patterns in human umbilical vein endothelial cells (HUVECs) from a diverse population of newborns.

### 3.4. How Could Epigenetic Clocks Support Treatment Decisions in Kidney Disease

Epigenetic clocks offer insights into kidney-specific aging processes; they may significantly impact treatment decisions by addressing the rate of disease-progression, the follow-up, and the screening process.

Specifically, by comparing a patient’s biological kidney age to their chronological age, clinicians could tailor interventions. Patients showing accelerated kidney aging might benefit from more aggressive therapies, preventing progression to end-stage renal disease, while those with slower aging may require less intensive treatments.

Epigenetic clocks can also serve as powerful tools for monitoring disease progression. By tracking changes in biological kidney age, clinicians can evaluate treatment efficacy and predict disease trajectory. This dynamic approach may enable timely adjustments to care plans.

Moreover, these clocks may identify individuals at high risk of developing kidney disease, facilitating early interventions and potentially preventing ESRD. By uncovering the mechanisms underlying kidney disease, such as inflammation or fibrosis, epigenetic research may inform the development of targeted therapies and biomarkers.

Implementing epigenetic clocks in clinical practice requires standardized protocols, clinician training, and regulatory approval. While challenges exist, such as establishing reliable measurement methods and addressing ethical considerations, the potential benefits are immense. As research advances, we anticipate the widespread adoption of epigenetic clocks, leading to more effective, personalized kidney disease management and improved patient outcomes.

Specific examples of applications of epigenetic clocks in personalized medicine include patient frailty estimates [16], anti-aging compound testing [63,64], and the modulation of drugs able to accelerate (e.g., NSAID analgesics [65]) or slow down (e.g., calcium channel blockers [65], or the combination of growth hormone, dehydroepiandrosterone, and metformin [66]) epigenetic clocks. The integration of epigenetic clocks and microbiome profiles into personalized medicine may change current practices. This innovative method expands beyond traditional biomarkers to provide a method for measuring lifestyle and environmental factors influencing gene expression and microbial communities. The major barriers to applying the proposed approach in clinical practice reside in the costs and interpretation of the data. However, these limitations can be easily overcome once the major epigenetic sites and microbial species contributing to kidney health have been identified. Once the number of epigenetic sites and microbial species needing to be identified has been sufficiently restricted, the overall cost of the analysis will dramatically drop, thus allowing for a wide application.

## 4. Conclusions

Disease variability cannot be solely explained at the level of individual cells. It is a phenomenon that occurs at the level of the organism or in complex tissues. This requires the presence of a biological clock, possibly digital, that is also probably involved in the growth process. The primary digital clock we know to be associated with aging is the epigenetic clock, and this may be relevant to endothelial dysfunction and the progressive fibrosis of the kidney. Studying this clock could lead to the development of strategies to stop the premature aging process in IgAN.

The interplay between the gut microbiota and kidney health in IgAN is a promising area of research, offering new avenues for personalized treatment strategies. Insights from IBD highlight the potential use of microbiota-based interventions, including probiotics, prebiotics, FMT, and dietary modifications, in managing IgAN. Future research and clinical trials will be crucial in validating and integrating these approaches into standard care for IgAN patients, ultimately improving their prognosis and quality of life.

A personalized medicine approach based on microbiome and epigenome data might result in a personalized treatment strategy for kidney diseases.

## Figures and Tables

**Figure 1 ijms-25-08592-f001:**
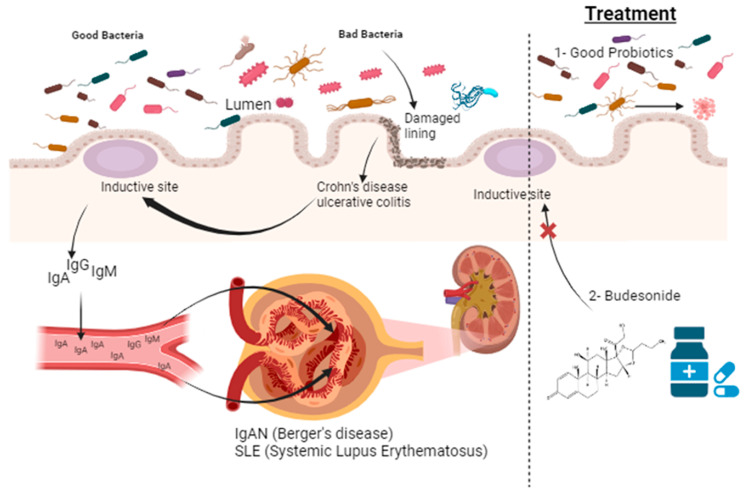
Gut microbiota and kidney disease in personalized medicine. Changes in gut microbiota are thought to induce an overactivation of the gut immune system. This may result in a local inflammatory disease (inflammatory bowel disease, such as Crohn’s disease or ulcerative colitis), or damage to other organs such as in Systemic Lupus Erythematosus (SLE) or IgA Nephropathy (IgAN). The regulation of gut dysbiosis using a personalized medicine approach may reduce the chance of these conditions being triggered. Arrows indicate stimulation, crossed arrow (X) means inhibition.

**Figure 2 ijms-25-08592-f002:**
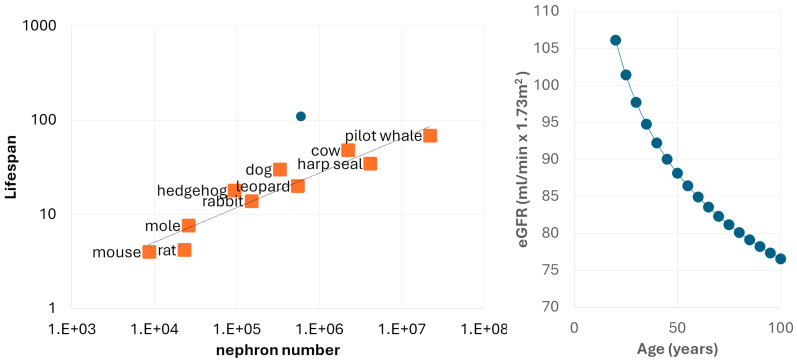
**Left**: Relationship between number of glomeruli and maximum lifespan. The blue dot represents Homo sapiens. **Right**: Graphical representation of the MDRD formula reported in the text, with theoretical relationship between age and eGFR (the constant value of the formula has been fixed considering a male Caucasian subject and a constant creatinine of 0.96). The number of glomeruli in different mammals has been derived from [51,52]. The lifespans of different mammals have been derived from [53] and from the AnAge database (http://genomics.senescence.info/species/ accessed on 10 June 2007) and the database from [54] (https://www.demogr.mpg.de/longevityrecords/0203.htm accessed on 10 June 2023).

**Figure 3 ijms-25-08592-f003:**
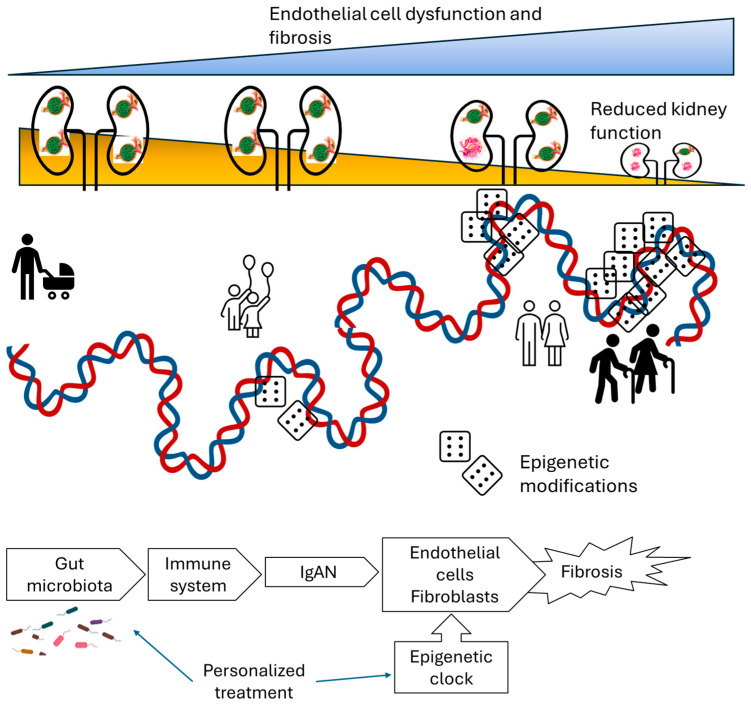
Epigenetic clock and kidney disease in personalized medicine. **Upper panel**: The epigenetic landscape of DNA is correlated with the human ageing process, and therefore with the individual variability in kidney diseases such as IgAN. It is possible that the ageing of endothelial cells mediates this relationship. **Lower panel**: Overall hypothesis regarding the microbiota and epigenetic clock control of kidney disease variability. Gut dysbiosis may represent a relevant trigger for kidney diseases such as IgAN. The epigenetic clock represents a modulator of endothelial and fibroblast responses to the immune trigger. A personalized medicine approach based on microbiome and epigenome data might result in a personalized treatment strategy.

**Table 1 ijms-25-08592-t001:** Gut bacteria associated with three typical autoimmune diseases: IgAN, SLE and Crohn’s disease.

Bacteria	IgAN	SLE	Crohn’s Disease
*Enterobacteriaceae*	Increased [9]		Increased [10]
*Proteobacteria*		Increased [11]	Increased [12]
*Firmicutes*		Decreased [11,13]	Decreased [12]
*Sutterellaceae*	Increased [9]		
*Bifidobacterium*	Decreased [9]		
*Lactobacillaceae*		Increased [13,14,15]	
*Moraxellaceae*		Increased [14]	
*Streptococcus*, *Megasphaera*, *Fusobacterium*, *Oribacterium*, *Odoribacter*, *Blautia*, and *Campylobacter*		Increased [15]	
*Enterococcus gallinarum*		Increased [13]	
Veilonella		Increased [14,15]	
*Corynebacteriaceae*, *Micrococcaceae*, *Defluviitaleaceae*, *Caulobacteraceae*, *Phyllobacteriaceae*, *Methylobacteriaceae*, *Hyphomicrobiaceae*, *Sphingomonadaceae*, *Halomonadaceae*, *Pseudomonadaceae*, *Xanthomonadaceae*		Decreased [14]	
*Bacteroidetes*		Decreased [11,13]	
*Faecalibacterium* and *Roseburia*		Decreased [15]	
*Enterobacteriaceae*	Increased [9]		Increased [10]
*Proteobacteria*		Increased [11]	Increased [12]
*Firmicutes*		Decreased [11,13]	Decreased [12]
*Sutterellaceae*	Increased [9]		
*Bifidobacterium*	Decreased [9]		
*Lactobacillaceae*		Increased [13,14,15]	
*Moraxellaceae*		Increased [14]	
*Streptococcus*, *Megasphaera*, *Fusobacterium*, *Oribacterium*, *Odoribacter*, *Blautia*, and *Campylobacter*		Increased [15]	
*Enterococcus gallinarum*		Increased [13]	
*Veilonella*		Increased [14,15]	
*Corynebacteriaceae*, *Micrococcaceae*, *Defluviitaleaceae*, *Caulobacteraceae*, *Phyllobacteriaceae*, *Methylobacteriaceae*, *Hyphomicrobiaceae*, *Sphingomonadaceae*, *Halomonadaceae*, *Pseudomonadaceae*, *Xanthomonadaceae*		Decreased [14]	
*Bacteroidetes*		Decreased [11,13]	
*Faecalibacterium* and *Roseburia*		Decreased [15]

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
