# Peer review of "Epigenetics, Microbiome and Personalized Medicine: Focus on Kidney Disease"

_ijms, 2024, doi:10.3390/ijms25168592_

Round 1

Reviewer 1 Report

Comments and Suggestions for Authors

Several studies showed that epigenetic clocks can play an important role in age measuring and in the prediction of diseases incidence and mortality. The present review may contribute to a better understanding of the role of microbiome in the onset and progression of different pathologies, including renal diseases. Even if the data are well described and the conclusions are supported by the presented information, there some minor suggestions that should be taken into consideration:

- the data presented in Figure 2 should be better described - more information of the study that noticed these results; in addition, it is not clear who conducted this research. Please, be sure you have copyright permission for this figure and also the image is not clear enough (low pixel quality).

- the conclusion section should not include additional figures; it should only present the conclusions of the authors. Please, delete Figure 4 from the conclusions and include it perhaps in section 3.2.

Author Response

Several studies showed that epigenetic clocks can play an important role in age measuring and in the prediction of diseases incidence and mortality. The present review may contribute to a better understanding of the role of microbiome in the onset and progression of different pathologies, including renal diseases. Even if the data are well described and the conclusions are supported by the presented information, there some minor suggestions that should be taken into consideration: 

- the data presented in Figure 2 should be better described - more information of the study that noticed these results; in addition, it is not clear who conducted this research. Please, be sure you have copyright permission for this figure and also the image is not clear enough (low pixel quality). 

 Answer. The right part of Figure 2 is the graphical representation of the MDRD formula. Regarding the left panel of Figure 2, we have now included the requested information in the text as follows: 

“Left: Relationship between number of glomeruli and maximum lifespan. Right: graphical representation of the MDRD formula reported in the text, with theoretical relationship between age and eGFR (the constant value of the formula has been fixed considering a male Caucasian subject). The number of glomeruli in different mammals has been derived from refs [42,43]. Lifespan of different mammals has been derived from [44] and from the AnAge database (http://genomics.senescence.info/species/ June 2007) and the database from ref [45] (https://www.demogr.mpg.de/longevityrecords/0203.htm ).” 

- the conclusion section should not include additional figures; it should only present the conclusions of the authors. Please, delete Figure 4 from the conclusions and include it perhaps in section 3.2. 

Answer. Done. Figure 4 has been inserted in section 3.3 and merged with Figure 3.

Reviewer 2 Report

Comments and Suggestions for Authors

The manuscript presents a thought-provoking perspective on the potential of epigenetics and microbiome data for the personalized treatment of kidney disease. The topic is highly relevant and engaging, which makes the review interesting. Although the authors make some reasonable arguments for considering these multidimensional datasets as continuous variables to address the challenges of personalized medicine, the manuscript takes a strong theoretical approach that raises many questions. If I, as a reader, have these questions, other readers may have the same. So, I suggest the authors address the following questions in the text.

1)      How could epigenetic clocks support treatment decisions in kidney disease?

2)      What are some examples of personalized interventions that could be based on epigenetic clocks?

3)   How does the proposed approach to assessing epigenetic clocks and microbiome profiles differ from current approaches? Further explanation is needed to clarify how this impacts personalized medicine in practice.

4)      How can the proposed approach be implemented in clinical practice? What are the barriers?

5)      Why the MDRD but not the CKD-EPI equation was used?

6)      The abstract and conclusions focus on IgA nephropathy, but the main text does not discuss this disease in detail. In my opinion, greater clarity of scope and focus is needed. It would be helpful either to expand the examples of IgA nephropathy or to clarify that the principles apply more generally.

Comments on the Quality of English Language

There are a few grammatical and typographical errors that need correction.

Author Response

The manuscript presents a thought-provoking perspective on the potential of epigenetics and microbiome data for the personalized treatment of kidney disease. The topic is highly relevant and engaging, which makes the review interesting. Although the authors make some reasonable arguments for considering these multidimensional datasets as continuous variables to address the challenges of personalized medicine, the manuscript takes a strong theoretical approach that raises many questions. If I, as a reader, have these questions, other readers may have the same. So, I suggest the authors address the following questions in the text.  

  1. How could epigenetic clocks support treatment decisions in kidney disease?

Answer. We have now added a specific paragraph describing this topic as follows:

How could epigenetic clocks support treatment decisions in kidney disease. Epigenetic clocks offer insights into kidney-specific aging processes; they may significantly impact treatment decisions, by addressing the rate of disease-progression, the follow-up, and the screening process.

Specifically, by comparing a patient's biological kidney age to their chronological age, clinicians could tailor interventions. Patients with accelerated kidney aging might benefit from more aggressive therapies to prevent progression to end-stage renal disease, while those with slower aging may require less intensive treatments.

Epigenetic clocks can also serve as powerful tools for monitoring disease progression. By tracking changes in biological kidney age, clinicians can evaluate treatment efficacy and predict disease trajectory. This dynamic approach may enable timely adjustments to care plans.

Moreover, these clocks may identify individuals at high risk of developing kidney disease, facilitating early interventions and potentially preventing ESRD. By uncovering underlying mechanisms of kidney disease, such as inflammation or fibrosis, epigenetic research may inform the development of targeted therapies and biomarkers.

Implementing epigenetic clocks in clinical practice requires standardized protocols, clinician training, and regulatory approval. While challenges exist, such as establishing reliable measurement methods and addressing ethical considerations, the potential benefits are immense. As research advances, we anticipate widespread adoption of epigenetic clocks, leading to more effective, personalized kidney disease management and improved patient outcomes.

  1. What are some examples of personalized interventions that could be based on epigenetic clocks?

Answer. We have added the following description:

 “Specific examples of applications of epigenetic clocks in personalized medicine are the patient frailty estimates [16], anti-aging compound testing [54,55] , modulation of drugs able to accelerate (e.g. NSAID analgesics [56] ) or slow down (e.g. calcium channel blockers [56], or the combination of growth hormone, dehydroepiandrosterone, and metformin [57]) epigenetic clocks.”

  1. How does the proposed approach to assessing epigenetic clocks and microbiome profiles differ from current approaches? Further explanation is needed to clarify how this impacts personalized medicine in practice.

Answer. We have now added the following explanation: “The integration of epigenetic clocks and microbiome profiles into personalized medicine may change current practices. This innovative method expands beyond traditional biomarkers to provide a method to measure and influence lifestyle and environmental factors on gene expression and microbial communities.

  1. How can the proposed approach be implemented in clinical practice? What are the barriers?

Answer. We have now inserted the following explanation: “The major barriers to applying the proposed approach in clinical practice reside in the costs and interpretation of the data. However, these limitations can be easily overcome once the major epigenetic sites and microbial species contributing to kidney health are identified. Once the number of epigenetic sites and microbial species needing to be identified restricted enough, the overall cost of the analysis will dramatically drop down, thus allowing for a wide application.”

  1. Why the MDRD but not the CKD-EPI equation was used?

Answer. We explain now this choice as follows:

“Here we have used the MDRD equation rather than other well established estimates of GFR, such as the CKD-EPI because it has a smooth, continuous behavior. The MDRD formula is:

eGFR = k1 * SCr k2 * Age k3

where SCr is serum creatinine, k1, K2 and K3 are three constants (K1 depends on gender and race).

Conversely, the CKD-EPI formula is:

eGFR = k1 * min (SCr/k2, 1)k3 * max (SCr/k2, 1)k4 * K5Age

the reader can easily identify the two non-linear function “min” and “max” which introduce a non-smooth behavior of the function.

Though CKD-EPI equation shows an improvement in the estimate of the average eGFR in a population, there little if any improvement in the variability of the estimates around the mean; the improvement of CKD-EPI regarding the estimate of the true average GFR in a population comes at the cost of greater complexity and non-linearity of the formula, which might represent a major theoretical limitation for the generalizability of data to populations not included initially in the CKD-EPI study. These considerations led us to adopt the MDRD formula in the present study.”

  1. The abstract and conclusions focus on IgA nephropathy, but the main text does not discuss this disease in detail. In my opinion, greater clarity of scope and focus is needed. It would be helpful either to expand the examples of IgA nephropathy or to clarify that the principles apply more generally.

Answer. We now discuss the IgA nephropathy in more detail as follows: 

“In recent years, the complex interplay between IgA nephropathy (IgAN), gut microbiota, and epigenetic aging has emerged as a promising area of research. IgAN, characterized by IgA immune complex deposition in the kidneys, appears to be influenced by gut dysbiosis, which can disrupt intestinal permeability and trigger abnormal immune responses. The gut microbiome's role in modulating immune function and producing beneficial metabolites, such as short-chain fatty acids, suggests potential therapeutic avenues for IgAN management. Concurrently, epigenetic aging, measured by DNA methylation patterns, may offer insights into disease progression and severity in IgAN patients. Chronic inflammation, a hallmark of IgAN, can induce epigenetic modifications that may accelerate biological aging. Environmental factors, including diet and lifestyle, further complicate this relationship by influencing the microbiome and epigenome. As our understanding of these interconnected processes grows, novel approaches to IgAN treatment are emerging. These include microbiome modulation strategies, developing epigenetic biomarkers for improved diagnosis and prognosis, and exploring anti-aging interventions to mitigate disease progression. Ultimately, a personalized medicine approach that integrates microbiome profiles, epigenetic measurements, and traditional clinical markers may pave the way for more effective, targeted treatments for IgAN patients.” 

Reviewer 3 Report

Comments and Suggestions for Authors

Major remarks

1. The roles of gut microbiota and epigenetics are discussed as two separate parts, without any links between them. Therefore, I don't understand why these two topics were combined in one article.

2. The article is quite superficial, I would suggest to discuss the issue more profoundly.

3. It is unknown what were criteria for literature choice.

Minor remarks

1. Lack of citations for lines 125-155.

2. What does the blue dot mean at the left panel of Fig. 2?

Author Response

Major remarks 

  1. The roles of gut microbiota and epigenetics are discussed as two separate parts, without any links between them. Therefore, I don't understand why these two topics were combined in one article.

 Answer. We have now inserted a new paragraph about microbiota and epigenetics to explain the link as follows:

“Recent advances in biomedical research have revealed a fascinating and complex relationship between epigenetics and the gut microbiota, two fields that were once considered distinct but are now recognized as profoundly interconnected. This intricate interplay has significant implications for human health and disease, opening new avenues for therapeutic interventions and personalized medicine.

Epigenetics, the study of heritable changes in gene expression that do not involve alterations to the underlying DNA sequence, has emerged as a critical factor in understanding how environmental influences can affect gene activity. Concurrently, the microbiota, particularly the diverse community of microorganisms residing in the gut, has been recognized as a major player in human physiology, influencing everything from digestion and immunity to brain function.

The bidirectional communication between epigenetics and the microbiota forms a dynamic system that shapes various aspects of human health. On one hand, the microbiota is an environmental factor that influences epigenetic modifications. Bacterial metabolites, such as short-chain fatty acids (SCFAs), have been shown to affect epigenetic markers like DNA methylation and histone acetylation directly. These modifications can, in turn, alter gene expression patterns in host cells, potentially influencing susceptibility to various diseases.

Conversely, epigenetic mechanisms can significantly impact the composition and function of the gut microbiota. For instance, epigenetic modifications can modulate the expression of genes involved in maintaining the intestinal barrier function, thereby influencing the types of bacteria that colonize the gut. This reciprocal relationship creates a complex feedback loop, where changes in one system can cascade through the other, affecting overall health status.

The implications of this interplay extend to various health conditions and diseases. In the context of gut health, dysbiosis – an imbalance in the gut microbiota – has been linked to inflammatory diseases such as inflammatory bowel disease (IBD) and irritable bowel syndrome (IBS). Epigenetic changes influenced by the microbiota may contribute to the development and progression of these conditions. Similarly, in metabolic disorders like obesity and type 2 diabetes, the gut microbiota's role in energy metabolism interacts with epigenetic modifications, potentially influencing disease onset and progression.

Furthermore, the gut-brain axis, a complex network of bidirectional communication between the gut and the brain, exemplifies how the microbiota can influence brain function through epigenetic mechanisms. This interaction impacts mental health, impacting mood, behavior, and cognitive function.

  1. The article is quite superficial, I would suggest to discuss the issue more profoundly.

Answer. We have now inserted many new paragraphs that analyze in more depth several aspects discussed.

  1. It is unknown what were criteria for literature choice.

Answer. “This is a narrative review of the combined use of microbiota and epigenetics as tools in personalized medicine. Because the literature is scanty, it was impossible to use a rigorous analysis of the literature as in meta-analysis works, which will be possible only when larger amounts of clinical data are available. Notwithstanding, the new area of research has exciting potential for future therapeutic impact, and the authors believe having a starting review is of great help to the clinical and scientific community.”

Minor remarks 

  1. Lack of citations for lines 125-155.

Answer. We have now inserted the citations as follow: 

“IgA nephropathy (IgAN) is a chronic kidney disease characterized by the deposition of IgA antibodies in the glomeruli, leading to inflammation and progressive renal damage [17] . The exact etiology of IgAN is not fully understood, but recent research suggests that gut microbiota may play a significant role [18]. Accordingly, a major hypothesis is that aberrant immune responses to gut mucosal antigens might foster the production of an IgA variant that, released in the bloodstream, is entrapped among the mesangial cells of the kidney glomeruli. Obviously, this hypothesis supports the view of a change in the gut mi-crobiota as a major contribution to the stimulation of the gut immune system (see Figure 1)[19].

The imbalance of the gut microbiota or dysbiosis would alter mucosal immunity and overproduction of aberrantly glycosylated IgA, which may than deposit in the glomeruli.

The mechanism linking gut dysbiosis and aberrant IgA production is, however, not well defined. It has been hypothesized that specifically “toxic” bacteria may disrupt the gut barrier and gap junctions, which are very relevant in many organs, and thus increase intestinal permeability, allowing for the adsorption of “pro-inflammatory” microbial products. It is also possible that specific bacteria induce a chronic gut inflammatory state, as in IBD [20].

IBD encompasses Crohn's disease and ulcerative colitis, both showing chronic in-flammation of the gastrointestinal tract. Being an inflammatory state of the digestive sys-tem, this group of diseases is strongly suspected to be driven by gut dysbiosis [21]. Some interventions support this hypothesis: 1) some probiotics (living “protective” microorgan-isms ingested as foods or supplements) have shown promise in IBD, particularly strains like Lactobacillus and Bifidobacterium [22]; 2) fecal microbiota transplantation (transfer of fecal bacteria from a healthy donor to a patient) can induce IBD remission [23]; 3) FOD-MAP diet and exclusive enteral nutrition have been effective in IBD [24]; 4) antibiotics that can disrupt gut microbiota may reduce inflammation in IBD [25]. “

  1. What does the blue dot mean at the left panel of Fig. 2?

 Answ. we have inserted the description in the figure legend as follows: “The blue dot represents Homo sapiens.”

Round 2

Reviewer 3 Report

Comments and Suggestions for Authors

Thank you for making the proposed improvements. The article is OK now. My only remark is that the changes made shoud be highlighted in the manuscript, it would ease the assessment of the revised version.